# High-Throughput Chlorophyll and Carotenoid Profiling Reveals Positive Associations with Sugar and Apocarotenoid Volatile Content in Fruits of Tomato Varieties in Modern and Wild Accessions

**DOI:** 10.3390/metabo11060398

**Published:** 2021-06-18

**Authors:** Yusuke Aono, Yonathan Asikin, Ning Wang, Denise Tieman, Harry Klee, Miyako Kusano

**Affiliations:** 1Degree Programs in Life and Earth Sciences, University of Tsukuba, Tsukuba 305-8572, Ibaraki, Japan; s2030229@s.tsukuba.ac.jp; 2Department of Bioscience and Biotechnology, Faculty of Agriculture, University of the Ryukyus, Nishihara 903-0213, Okinawa, Japan; y-asikin@agr.u-ryukyu.ac.jp; 3Faculty of Life and Environmental Science, University of Tsukuba, Tsukuba 305-8572, Ibaraki, Japan; wang.ning.fu@u.tsukuba.ac.jp; 4Tsukuba-Plant Innovation Research Center, University of Tsukuba, Tsukuba 305-8572, Ibaraki, Japan; 5Department of Horticultural Sciences, University of Florida, Gainesville, FL 32611, USA; dtieman@ufl.edu (D.T.); hjklee@ufl.edu (H.K.); 6RIKEN Center for Sustainable Resource Science, Yokohama 230-0045, Kanagawa, Japan

**Keywords:** tomato, fruit pigment, flavor, apocarotenoid, sugar accumulation, aroma compounds, carotenoid, lycopene, chlorophyll

## Abstract

Flavor and nutritional quality has been negatively impacted during the course of domestication and improvement of the cultivated tomato (*Solanum* *lycopersicum*). Recent emphasis on consumers has emphasized breeding strategies that focus on flavor-associated chemicals, including sugars, acids, and aroma compounds. Carotenoids indirectly affect flavor as precursors of aroma compounds, while chlorophylls contribute to sugar production through photosynthesis. However, the relationships between these pigments and flavor content are still unclear. In this study, we developed a simple and high-throughput method to quantify chlorophylls and carotenoids. This method was applied to over one hundred tomato varieties, including *S*. *lycopersicum* and its wild relatives (*S*. *l*. var. *cerasiforme* and *S*. *pimpinellifolium*), for quantification of these pigments in fruits. The results obtained by integrating data of the pigments, soluble solids, sugars, and aroma compounds indicate that (i) chlorophyll-abundant varieties have relatively higher sugar accumulations and (ii) prolycopene is associated with an abundance of linear carotenoid-derived aroma compounds in one of the orange-fruited varieties, “Dixie Golden Giant”. Our results suggest the importance of these pigments not only as components of fruit color but also as factors influencing flavor traits, such as sugars and aroma.

## 1. Introduction

Tomato is one of the most important vegetables worldwide. Recent breeding strategies have focused not only producer-favorable traits such as high yield, long shelf life, and disease resistance, but also on flavor [1]. The major flavors of tomato are the result of interactions between taste and aroma [2]. In tomato fruits, sugars (mainly glucose and fructose) and acids (mainly citrate, malate, and ascorbate) are perceived by taste receptors, while various volatile organic compounds (VOCs) are perceived by olfactory receptors [3]. To improve fruit flavors, many genetic approaches have been conducted [4,5,6]. Wild relatives that can be crossed with cultivated tomatoes (*S. lycopersicum*), such as *S. l.* var. *cerasiforme* and *S*. *pimpinellifolium*, are important genetic resources that have been exploited to provide insights into the synthesis of flavor-associated chemicals [5,6].

Fruit color is also an important factor in determining commercial value [7]. Among cultivated tomatoes (*S**. lycopersicum*), several varieties show various colors of mature fruits, including green, orange, pink, brown, yellow, and red [8]. Fruit color in tomato is produced by several pigments. The pericarp contains carotenoids and chlorophylls, while the peel primarily contains flavonoids [9]. In parallel with the change from green to red during ripening, various events, such as the accumulation of carotenoids, degradation of chlorophylls, starch decomposition into monosaccharides, reduction of organic acids, and fruit softening, occur in a coordinated manner [10,11]. The ripening process also causes changes in VOC composition [12]. VOCs at the mature red stage provide a commercially valuable aroma [13,14].

Prior research indicates that some tomato fruit color mutations can affect the composition of nutritional and flavor compounds; therefore, chlorophylls and carotenoids exhibit the potential to improve flavors in tomato fruits [15]. Even though photosynthetic capacity in tomato fruits is limited compared to leaves, chlorophyll content can be associated with sugar accumulation in fruits [16,17]. Carotenoids are known to be important for human health benefit to decrease the risk of disease [18]. Furthermore, carotenoids are precursors of apocarotenoid (AC)-VOCs, providing fruity/floral aroma notes, although carotenoids themselves are tasteless [19]. AC-VOCs can enhance the perception of sweetness [20] and contribute positively to consumer preferences [21]. Several genes regulating fruit color and pigment content have been identified [22]. To date, the relationships between these pigments and flavors are not fully understood.

High-performance liquid chromatography (HPLC) combined with a photodiode array detector (PDA) is widely used for the identification and quantification of carotenoids [23,24,25,26]. However, these methods generally require 20 min or more per sample. This time commitment per sample is a burden on the researcher; therefore, a higher-throughput method is highly desirable for analysis of large sample sets. For quantification of carotenoids and chlorophylls, simple methods using a spectrophotometer are well established [27,28,29]. Despite the acceptance of these methods, optimization is still needed to achieve high-throughput measurements that will enable the analysis of large sample populations.

In this study, we developed a simple and rapid method for the quantification of non-polar pigments. Applying this method, we quantified these pigments in mature fruits of over one hundred varieties of tomato. Combining this pigment analysis with flavor analysis suggested that the sugar content can be linked to the accumulation of chlorophylls in tomato fruit. Additionally, one tomato variety in this analysis showed a unique carotenoid metabolite composition, including increased AC-VOCs.

## 2. Results

### 2.1. Method Development for High-Throughput Quantification of Carotenoids, Lycopene, and Chlorophylls in Tomato Fruits

To provide new insights into the relationship between nonpolar pigments and flavor-related compounds, we developed the simple and high-throughput method for pigment quantification in tomato fruits (Figure 1). When comparing classical methods using a spectrophotometer with a spectrophotometer cell, the developed method using 96-well plates and a microplate reader exhibited three distinct advantages. First, sample measurements require only 300 μL, equivalent to only 0.75 mg of dry sample weight. Second, the use of a multimode microplate reader to measure five wavelengths allowed for the quantification of carotenoids, lycopene, and chlorophylls a and b simultaneously. The developed method can quantify the total content of carotenoids and lycopene content at the wavelengths of 470 and 506 nm, respectively. The limits of detection of each analyte (chlorophyll a, chlorophyll b, total carotenoids, and lycopene) were 0.029, 0.12, 0.05, and 0.071 μg/mL, respectively (Appendix A). The sensitivity for carotenoid and lycopene quantification was comparable to previous HPLC analysis [30]. The accuracy of the method was validated by comparing the results of the pigment content in “Micro-Tom” fruits obtained from HPLC-PDA analysis (Appendix A). The results of the pigment content analyzed by the microplate reader and by HPLC-PDA showed a positive correlation with a high correlation coefficient (R^2^ > 0.8). Third, the acquisition time is very short; each sample is analyzed in approximately 10 s. The method can analyze 24 samples per 2.5 min automatically as one batch. These results suggest that the developed method allows the quantification of hundreds of samples much faster than the use of HPLC-PDA methods with equivalent accuracy.

AC-VOCs are known to accumulate during ripening [14]. In the case of tomato, the ripening stage is discriminated based on skin color appearance [31]. It is well-known that skin colors in fruits, including tomato, are associated with chlorophyll content during ripening [32]. In most tomato varieties, skin color turns from green to red. The developed method can capture common alternations of pigment content in each stage of tomato fruit ripening in well-studied varieties, i.e., “Ailsa Craig” (AC), “Moneymaker” (MM), and “Micro-Tom” (MT), at the three different ripening stages (Figure 2 and Appendix A). To obtain insights into the accumulation of pigments and flavor-related compounds such as sugars and AC-VOCs, we applied the method to quantify the pigments in 157 varieties of tomato fruits.

### 2.2. Pigment-Associated Characterization of 157 Varieties of Tomato Fruit

Results of the pigment quantification of 157 varieties are shown in Figure 3 and Appendix A. The same “Moneymaker” RR fruit was analyzed in each batch as quality control to validate technical errors in analytical replicates (Appendix A). Similar to AC, MT, and MM (Figure 2d–f), lycopene content accounted for over 50% of total carotenoids in the fruits of 142 of 157 varieties. Conversely, 12 varieties contained 40% or less lycopene (Appendix A). This result suggested that the composition of carotenoids, especially the ratio of lycopene to other carotenoids, is widely conserved in lycopene-accumulating varieties, such as red- or brown-fruited varieties, even though the total amount of carotenoids varied. This trend was not observed in yellow- and green-fruited varieties that showed no accumulation of lycopene.

### 2.3. Association between Pigments and Flavor-Related Compounds

To elucidate the effect of fruit color-based selection on flavor, we compared quantitative data of the pigments with those of flavor-related compounds from [21], as taste-related compounds, soluble solids, glucose, and fructose content are available for all varieties except for “Bear Creek.” Chlorophylls are the primary pigments and vital for photosynthesis, while sugars can be synthesized via the Calvin–Benson cycle. To obtain insights between chlorophyll content and sugar accumulation in tomato fruits, we evaluated the association between them (Figure 4a,b and Appendix A). The average of soluble solids, glucose, and fructose content in the 16 varieties with the top 10% chlorophyll a content was significantly higher than that of the bottom 10%. Of the 16 varieties, three varieties were MG-like or BR-like (proportion of chlorophyll content was >50% in total pigments with less lycopene), while the other 13 varieties were red-fruited types (lycopene ratio in total carotenoids was about 50%). This finding suggested an association between the amount of chlorophylls and the three taste-related metabolites, including sugars.

As several types of aroma compounds, such as AC-VOCs, can be synthesized by cleavage of carotenoids, we focused on two AC-VOCs, 6-methyl-5-hepten-2-one (MHO) and geranylacetone, in the 157 varieties [21]. In the 16 varieties with the bottom 10% total carotenoids content, eight varieties contained 1/10 or less of the average content of AC-VOCs (Figure 4c,d). Conversely, two varieties showed a very high AC-VOC accumulation despite very low lycopene and total carotenoid content. One is an orange-fruited variety named “Dixie Golden Giant,” and the other is a yellow-fruited variety named “Kellogg’s Breakfast” (Appendix A). In the two varieties, the MHO content, produced directly by the cleavage of lycopene, was about five times higher than the average. This finding suggested that a different carotenoid as a substrate to produce MHO in “Dixie Golden Giant” and “Kellogg’s Breakfast” likely exists.

### 2.4. Evaluation of Relationships between Fruit Color Variation, Carotenoid Composition, and AC-VOC Content in Eight Representative Varieties

We conducted HPLC-based carotenoid profiling to compare carotenoid accumulation patterns in eight tomato varieties. One was a red-fruited variety, and the other seven were chosen based on the pigment content and fruit color in the 157 varieties (Appendix A). The variety “Ailsa Craig” was used as a representative of a red-fruited tomato, while five varieties were chosen in *S*. *lycopersicum*. Two varieties (“Bear Creek” and “Chocolate Cherry”) showed a brown color. The variety “Dixie Golden Giant” exhibits an orange color skin and contains very high levels of AC-VOCs in fruits. The varieties “N135 Green Gage” and “Green Zebra” showed a BR-like color (yellow) and a MG-like color (green), respectively. Two *S*. *lycopersicum* var. *cerasiforme* varieties (“Lemon Drop” and “Poire Jaune”) showing yellow colors were also chosen. These varieties accumulated very low AC-VOC content in their fruits [21]. Chromatograms of these varieties are shown in Figure 5. As expected, lycopene-accumulating varieties, i.e., “Ailsa Craig” and the two varieties with a brown color, showed similar chromatographic patterns (Figure 5i–iii, Appendix A). Most of the peaks except for *beta*-carotene and lutein were missing in chromatograms of the four varieties that contained very low levels of lycopene (Figure 5v–viii, Appendix A). In the variety “Dixie Golden Giant,” an unknown peak (peak No. 9) was detected, and there was no accumulation of lycopene or other peaks found in the other varieties (Figure 5iv, Appendix A). Based on the UV spectrum pattern of the unknown compound and those of the reported carotenoids, the compound was presumed to be prolycopene (Appendix A) [23,25,33]. As the fruit over-accumulates MHO, prolycopene is a potential substrate to produce MHO instead of *trans*-lycopene (see Section 3).

## 3. Discussion

To investigate whether the quantity of the major nonpolar pigments in tomato fruits, i.e., carotenoids, lycopene, and chlorophylls, can affect the accumulation of flavor-related compounds, we developed a simple and high-throughput method for the quantification of these pigments. The developed method could capture changes in pigment composition during tomato ripening stages in three well-known varieties. The high-throughput quantification of these pigments in 157 tomato varieties, and the concentration of sugars and AC-VOCs enabled us to select eight varieties representing the range of color in ripe fruits for carotenoid profiling. The result showed that carotenoid profiles reflect the appearance of fruit color and AC-VOC accumulation. Among the tested varieties, the orange-fruited variety showed over-accumulation of prolycopene and MHO.

We integrated data of chlorophylls and taste-related compounds in the 157 tomato fruits. Our results showed that high sugar content was found in varieties with high chlorophyll accumulation (Figure 4a,b). This result suggests a link between chlorophylls and high sugar content. A candidate gene involving a regulatory link between chlorophylls and sugar production is *SlGLK2* [33,34]. This gene encodes a transcription factor affecting chloroplast development and localization. A functional allele causes a phenotype called “green shoulder,” showing a dark-green region on the stem end in green fruit found in many wild species. As it causes uneven ripening and often remains orange, green, or white in ripe fruit, the loss-of-function allele called “u” has been selected by breeders in almost all modern commercial cultivars. However, it is believed that the selection of this phenotype caused a reduction in sugar content in modern varieties. Thus far, this theory is based on analysis using near-isogenic wild type and *u* mutants. Our results demonstrate that chlorophylls in fruit positively correlate with sugar content in the actual variety population.

Our results indicate that lycopene content accounts for about half of the total carotenoids in fruit in approximately 90% (142/157) of the tomato varieties (Appendix A). In red tomato fruits, lycopene and its derivatives such as *beta*-carotene, lutein, and zeaxanthin are the main carotenoids present [35]. Our results suggest that the ratio of lycopene to these other carotenoids is widely preserved among varieties. One possible explanation is that carotenoid biosynthesis in tomato fruit can be tightly regulated by the expression balance between *PSY1* (a gene encoding a key enzyme in carotenoid synthesis) and genes involved in the downstream pathway of carotenoid biosynthesis [31,35]. Previous studies indicated that enhancement of the carotenoid pathway by over-expression of *PSY1* caused the feed-forward up-regulation of *CYC-B* transcription, a fruit-specific lycopene cyclase and lycopene cyclization activity [36]. In addition, over 30 regulatory elements are present in the promoter region of *PSY1* [37], potentially diversifying the regulation of expression. In summary, the total amount of carotenoids in tomato fruits can vary depending on *PSY1* expression, but the ratio of lycopene among the varieties may remain constant by controlling downstream pathways. It is known that carotenoid content in tomato fruit is largely influenced by changes in growth conditions (e.g., light, temperature, season, drought stress, salinity stress, and pedoclimatic conditions) [9,38,39,40,41]. Analyzing associations between carotenoid and AC-VOC content in tomato fruits grown under different growth conditions may reveal further clues for improving AC-VOC production.

When integrating carotenoid and aroma-related profiles, the variety ‘Dixie Golden Giant’ displayed an abnormal accumulation of prolycopene and MHO content. As these AC-VOCs can be generated by cleavage of the linear end of carotenoids [42], they can be derived from the all-*trans* form (all-*trans*-lycopene) or from the *cis* form (prolycopene). A recent study demonstrated that this variety exhibits a mutation in the gene coding for carotene isomerase (CRTISO), an enzyme involved in the isomerization of prolycopene (tetra-*cis*-lycopene) to all-*trans*-lycopene [43]. Generally, most lycopene in tomato fruit exists as the all-*trans* form and crystallizes in the tomato chromoplast [44]. Recent studies have shown that *cis*-carotenoids exhibit lower crystallinity and higher solubility than the all-*trans* form [45,46]. Researchers have shown that the loss of CRTISO function caused the over-accumulation of prolycopene instead of lycopene and its derivatives in fruit [47]. Additionally, mutants with a loss of CRTISO function display several times higher MHO and geranylacetone than their predecessors [20,48]. Similar to “Dixie Golden Giant,” “Kellogg’s Breakfast” also showed the highest MHO content (Figure 4). A previous study showed that the fruit color of the variety was yellow and the variety accumulated high prolycopene [49]. Given the over-accumulation of prolycopene in both varieties, its likely fate is to be efficiently converted to MHO and to contribute to the increase in tomato fruits.

In this study, we developed a high-throughput quantification method for nonpolar pigment profiling using a multimode microplate reader. Integrating data of pigment profiles, sugars, and AC-VOCs showed that (i) chlorophylls in fruit correlate with high sugar accumulation, and (ii) one of the yellow fruit varieties with high AC-VOC content, “Dixie Golden Giant,” accumulates prolycopene as a putative substrate.

In conclusion, our results suggest that carotenoids and chlorophylls in tomato can influence flavor traits, such as sugar content and aroma composition. The quantitative methods using a microplate reader can be a powerful tool to screen large-scale populations, such as mutant collections and natural accessions, not only for tomato but also for other fruits and vegetables accumulating these pigments.

## 4. Materials and Methods

### 4.1. Chemicals

All chemicals except for the standard of chlorophyll b, *trans*-β-apo-8′-Carotenal, and lutein were purchased from Fujifilm Wako Pure Chemical (Osaka, Japan). Standards of chlorophyll b and *trans*-*β*-apo-8′-carotenal were purchased from Sigma-Aldrich (Tokyo, Japan). Lutein standard was purchased from EXTRASYNTHESE S.A. (Genay, France).

### 4.2. Plant Materials and Sample Collection

Three varieties of tomato (*S. lycopersicum*) were grown, and the fruits were harvested at the University of Tsukuba. “Micro-Tom” plants were grown in a greenhouse under natural sunlight supplemented with additional light (16:30–22:00). Hyponex (Hyponex Japan Corp., Ltd., Osaka, Japan) diluted with water was applied to plants once a week as a nutrient. “Ailsa Craig” and “Moneymaker” plants were also grown in a greenhouse under natural sunlight. Hyponex (Hyponex Japan Corp., Ltd., Osaka, Japan) diluted with water was applied to plants once a week as a nutrient. Fruits of each variety were harvested at three different stages: (i) Mature Green (MG), (i) Breaker (BR), and (iii) Red Ripe (RR). The ripening stage of each fruit was estimated based on “USDA Visual Aid TM-L-1” (USDA Agri-cultural Marketing Service, 1975). Skin color of all harvested fruits was measured using a colorimeter (CM-700d, Konica Minolta, Tokyo, Japan). The average of three different points at the equatorial part of the fruit was calculated to represent the color of each fruit. For carotenoid measurement, fruits were lyophilized at least 48 h after chopping and removing seed and locular gel. After that, dried fruits were crushed and stored at −80 °C until analysis. In addition, 157 varieties of cultivated and wild tomatoes, including *S. lycopersicum*, *S. l.* var. *cerasiforme*, and *S. pimpinellifolium*, were cultivated and harvested at the University of Florida for screening. Harvested whole fruits were homogenized and lyophilized for at least 48 h and then stored at −80 °C until analysis.

### 4.3. Extraction and High-Throughput Measurement for Quantification of the Targeted Pigments Using Microplate Reader

Lyophilized fruits stored at −80 °C were acclimated to room temperature for 1 h. Two milligrams of lyophilized fruits were homogenized with 0.4 mL of acetone by TissueLyser II (Qiagen, Hilden, Germany) for 10 min at 15 Hz. After centrifugation (15,000 rpm, 5 min, 20 °C), the supernatant was collected and placed on ice. The same process was repeated twice, resulting in the collection of about 0.7 mL of extracts. For the absorbance measurement, a microplate reader (Infinite M200Pro, Tecan, Männedorf, Switzerland) was used. Three hundred microliters of the extracts were applied to a custom 96-well glass microplate (Nikkei Products CO., LTD, Osaka, Japan). The optical density was measured at 663, 647, 470, 506, and 750 nm (≤ ±1.5 nm). The following formula was used to correct the optical path length and calculate each wavelength:*Ax* = (*ODx*−*OD*_750_)/*l*(1)
(*Ax*: absorbance at *x* nm; *ODx*: measured optical density at *x* nm; *l*: path length of microplate (= 0.876887 cm)).

### 4.4. Calculation of Carotenoid and Chlorophyll Content

Using the formula created with reference to previous reports [27,28,29], chlorophyll a, chlorophyll b, total carotenoids, and lycopene content were calculated.
Chl a (mg/g DW): *Ca* = (11.24 *A*_662_−2.04 *A*_645_) × (*v*/*w*)(2)
Chl b (mg/g DW): *Cb* = (20.13 *A*_645_ − 4.19 *A*_662_) × (*v*/*w*)(3)
Total Carotenoids (mg/g DW) = {(1000 *A*_470_ − 1.90 *Ca* − 63.14 *Cb*)/214} × (*v*/*w*)(4)
Lycopene (mg/g DW) = (*A*_506_/315) × (*v*/*w*)(5)
*A*_x_: absorbance at *x* nm; *v*: volume of solvent (mL); *w*: weight of sample (mg)

### 4.5. Pigment Extraction and Carotenoid Profiling Using HPLC

For carotenoid extraction and HPLC analysis, we utilized a modification of Kimbara et al. [50]. After removal from storage, lyophilized fruit were acclimated to room temperature for 1 h. Seven milligrams of lyophilized fruit was extracted with 1155 μL of chloroform/methanol (1:1, *v*/*v*) containing 12 μM of *trans*-*β*-apo-8’-carotenal. The samples were homogenized by TissueLyser II (Qiagen, Hilden, Germany) for 5 min at 8 Hz. After adding 196 μL of water, the sample was centrifuged at 13,000× *g* for 10 min at room temperature. One thousand microliters of the supernatant was collected, and 240 μL of water was added. After keeping the sample on ice for 2 h and following centrifugation (1000× *g* for 10 min at 4 °C), 340 μL of the organic phase was collected. These extracts were concentrated by drying with SPD2010 SpeedVac^®^ (Thermo Fisher Scientific, Waltham, MA, USA) and resolving with 81 μL of chloroform/ethanol (1:1, *v*/*v*). Then, they were shaken for 5 min and sonicated for 2 min. After centrifugation (17,800× *g* for 15 min at 4 °C), 60 μL of supernatant was used for HPLC analysis.

An HPLC system (Shimadzu, Kyoto, Japan) was used in this analysis. One microliter of extract was injected into a Develosil C30-UG-5 2.0/250 (Nomura Chemical Co. Ltd., Seto, Japan) at a flow rate of 0.37 mL/min. The column oven was set to 30 °C. Eluent A, methanol containing 4.5% water and 0.1% triethylamine; and eluent B, hexane/isopropyl alcohol (60:40, *v*/*v*. 0.1% triethylamine), were used as the mobile phase. The gradient program was as shown in Appendix A. To detect analytes, a photodiode array detector (SPD-M20AD, Shimadzu, Kyoto, Japan) was used in the wavelength range between 200 and 800 nm. Chromatograms at 450 nm (for carotenoids) and 650 nm (for chlorophylls) were used to calculate the peak height. Peak identification and quantification were conducted using LabSolutions Ver. 5.89 (Shimadzu, Kyoto, Japan). Peaks were identified by their retention time and absorption spectra comparison to those of standards (chlorophyll a and b, *beta*-carotene, lutein, and lycopene) or estimated to be a carotenoid based on exhibiting absorption maxima at 400–500 nm [27].

### 4.6. Statistical Analysis

All quantitative values were calculated using Excel (Microsoft Office 2019). Welch’s *t*-test was performed by R 4.0.2. Values of *p* < 0.05 were considered significantly different.

## Figures and Tables

**Figure 1 metabolites-11-00398-f001:**
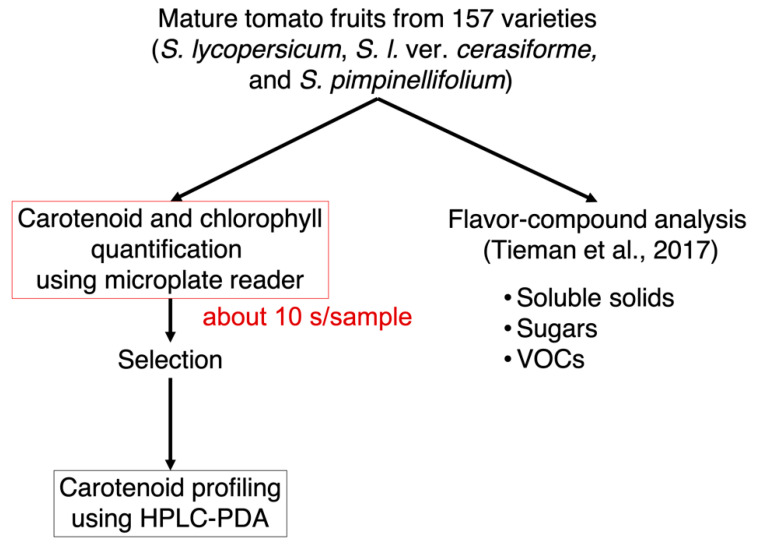
Workflow to quantify nonpolar pigments and flavor-related compounds in tomato fruits. The pigments (carotenoids, lycopene, and chlorophylls a and b) were first quantified by using a multimode microplate reader. After comparison of the pigment composition, several lines were chosen to conduct HPLC-PDA analysis for the quantification of three carotenoids, as well as chlorophylls. Data for flavor-related compounds such as soluble solids, sugars, and VOCs were obtained from Tieman et al. [21].

**Figure 2 metabolites-11-00398-f002:**
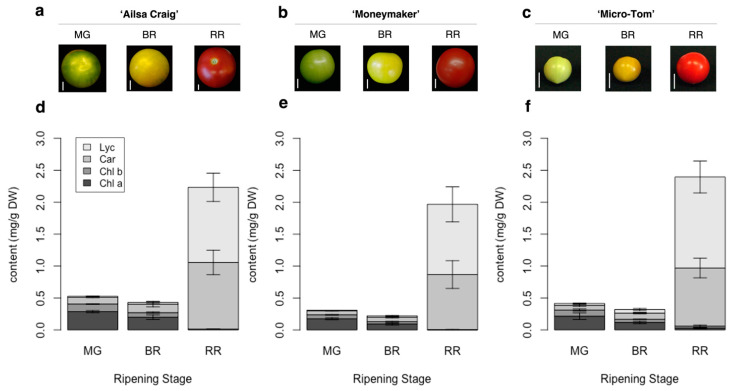
Skin color and pigment content in fruits of three tomato varieties during ripening. (**a**–**c**): Appearance and fruit size at different ripening stages of “Ailsa Craig,” “Moneymaker,” and “Micro-Tom.” The white bar indicates 1 cm. (**d**–**f**): Quantification of chlorophyll a (Chl a), chlorophyll b (Chl b), total carotenoids except for lycopene (Car), and lycopene (Lyc) content in the three varieties. Each value is shown as the average of biological replicates (*n* = 3) with standard deviation (SD).

**Figure 3 metabolites-11-00398-f003:**
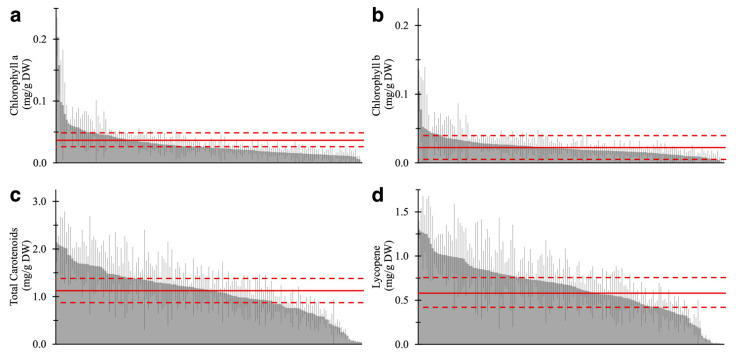
The four-pigment content in ripe fruit of 157 tomato varieties. (**a**) Chlorophyll a, (**b**) chlorophyll b, (**c**) total carotenoids, and (**d**) lycopene content are shown. The average and SD of the pigment content in AC are represented by the solid and dotted line, respectively. Each value is shown as the average of biological replicates (*n* = 2–3) and analytical replicates (*N* = 1–2) with SD.

**Figure 4 metabolites-11-00398-f004:**
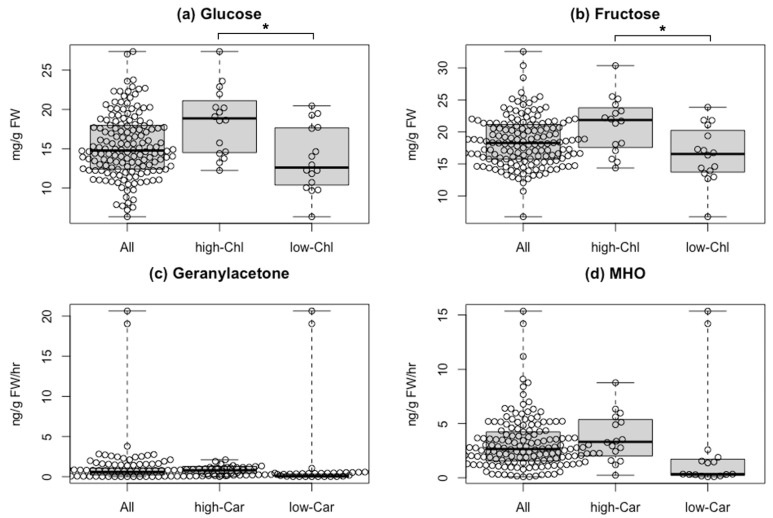
Comparison of flavor-related compounds between varieties with different pigment contents. (**a**) Glucose and (**b**) fructose in fruits of the 156 varieties (All), the top 16 varieties with high chlorophyll a content (high-Chl), and the bottom 16 varieties with low chlorophyll a content (low-Chl). Glucose and fructose contents in fruits of the 156 varieties were obtained from data in Tieman et al. [21]. (**c**) Geranylacetone and (**d**) MHO in fruits of the 157 varieties (All), the top 16 varieties with high total carotenoid content (high-Car), and the bottom 16 varieties with low total carotenoid content (low-Car). Geranylacetone and 6-methyl-5-hepten-2-one (MHO) contents in fruits of 157 varieties were obtained from Tieman et al. [21]. Asterisks represent a significant difference (* *p* < 0.05) assayed by Welch’s *t*-test. Abbreviation: FW, fresh weight.

**Figure 5 metabolites-11-00398-f005:**
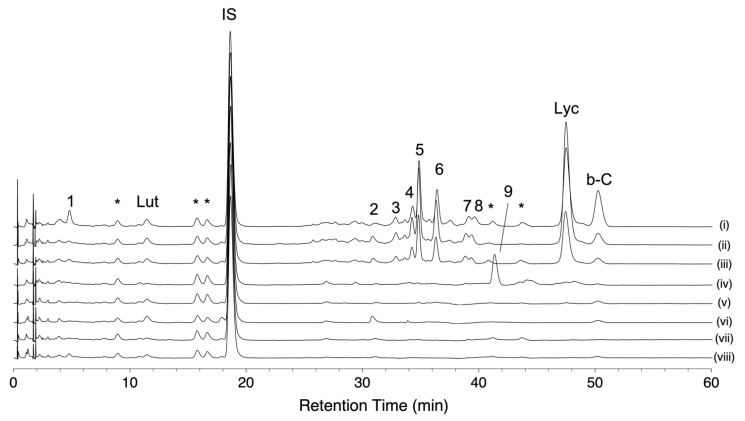
Carotenoid profiles of eight representative varieties analyzed by using HLPC-PDA. Carotenoid profiles were visualized at the wavelength of 450 nm. Each chromatogram shows the pattern of “Ailsa Craig” (**i**), “Bear Creek” (**ii**), “Chocolate Cherry” (**iii**), “Dixie Golden Giant” (**iv**), “Green Gage” (**v**), “Green Zebra” (**vi**), “Lemon Drop” (**vii**), and “Poire Jaune” (**viii**). Lycopene (Lyc), *beta*-carotene (b-C), and lutein (Lut) were identified by comparison to authentic standards. Spectral characteristics of peaks No. 1–9 are shown in Appendix A. Peaks with an asterisk are co-eluents that were not carotenoids (see Section 4 in detail).

## Data Availability

The data presented in this study are available in article and Appendix A.

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
