# Peer review of "High-Throughput Chlorophyll and Carotenoid Profiling Reveals Positive Associations with Sugar and Apocarotenoid Volatile Content in Fruits of Tomato Varieties in Modern and Wild Accessions"

_metabolites, 2021, doi:10.3390/metabo11060398_

Round 1
Reviewer 1 Report
The manuscript focused on carotenoid and chlorophyll pigment profiles of a wide range of tomato varieties, including accessions with contrasting fruit colour. The authors present an interesting high-throughput method for pigment quantification, requiring very low amount of sample and very little preparation. In addition, a correlation between chlorophyll levels and sugar content was outlined in tomato fruits, and a high level of apocarotenoid volatiles was detected in a variety which does not produce lycopene and was tentatively associated with the presence of prolycopene carotenoid.
I have a few minor concerns which should be addressed for manuscript acceptance:
- My main concern is about the technical error of the proposed method. I suggest adding information about how many replicates were used for pigment quantification, and what is the standard deviation of technical replicates?
For instance, in Figure S1 (calibration curves), error bars should be added in the plots.
In Figure 3, the standard deviation is very high for some varieties. Is this due to variation between biological replicates? Does the same SD exist when individual pigments are quantified by HPLC? Also, it is unclear how many replicates were used for HPLC analysis (is the deviation indicated in Tables S3 and S4 of technical or biological replicates?).
- Also, I am curious to know if the authors have a hypothesis about why the correlation between MPR and HPLC has a higher R2 for carotenoid content than for chlorophyll.
- Finally, I also wonder why was ‘Kellogg’s breakfast’ not included in the HPLC study (Evaluation of relationships between fruit color variation, carotenoid composition, and AC-VOC content)?
Is ‘Kellogg’s breakfast’ also accumulating high levels of the compound identified as prolycopene? Or is there another possible reason for its high levels of apocarotenoids.
A few more specific concerns:
Introduction
Lines 61-62: although carotenoids do not impact flavour, I would add a sentence or two describing the importance of these metabolites for the human diet (nutritional value).
Results
Table S1
Please correct legend:
Pigment content: is it the mean value? Of how many replicates?
Abbreviations: DW, dry weight.; % chl, % lyc/car, chlorophylls ratio in total pigments; % lyc/car, lycopene ratio in total carotenoids
In addition, %chl is not a ratio, so I suggest defining it as chlorophyll percentage in total pigments
The color of the ripe fruit of each variety should be included in Table S1.
Figure 3: Why was AC cultivar chosen as a reference? And what about MT and MM cultivars? They could be included in the plot, with an arrow indicating their respective position. Additionally, if the 157 varieties were measured ripe, I would remove the values for AC breaker stage (blue lines).
Line 158: specify Kellog’s Breakfast color
Discussion
Lines 230-232: the sentence is unclear. Please rewrite
Methods
Why was MicroTom grown with light supplementation?
Reviewer 2 Report
The work is interestingly written and deals with important issues. I have a suggestion to improve the work structure. In my opinion, the authors should clearly provide a summary of the most important theses at the end of the article or provide the most important conclusions. In this way, the reader can see the most important results of work in one place.
Reviewer 3 Report
The manuscript "High-throughput chlorophyll and carotenoid profiling reveals positive associations with sugar and apocarotenoid volatile
content in fruits of tomato varieties in modern and wild accessions "is an excellent work in all its parts, very well written.
The theme addressed by the authors is very important, especially because the mechanism of the extracting of the aromas is not yet clear.
The authors attempt to show that chlorophylls and carotenoids influence aromas, a very interesting theory, but in a study "A multivariate statistical analysis coming from the NMR metabolic profile of cherry tomatoes (The Sicilian Pachino case)" which must be mentioned, of shows that many characteristics are especially influenced by the pedoclimatic conditions and the production areas. So I suggest the authors to specify this in the text, as fruit under stressful conditions has been shown to produce more carotenoids.
